# A Rat Model of Post-Traumatic Stress Syndrome Causes Phenotype-Associated Morphological Changes and Hypofunction of the Adrenal Gland

**DOI:** 10.3390/ijms222413235

**Published:** 2021-12-08

**Authors:** Vadim Tseilikman, Maria Komelkova, Marina V. Kondashevskaya, Eugenia Manukhina, H. Fred Downey, Valerii Chereshnev, Margarita Chereshneva, Pavel Platkovskii, Anna Goryacheva, Anton Pashkov, Julia Fedotova, Olga Tseilikman, Natalya Maltseva, Olga Cherkasova, Charlotte Steenblock, Stefan R. Bornstein, Barbara Ettrich, George P. Chrousos, Enrico Ullmann

**Affiliations:** 1School of Medical Biology, South Ural State University, 454080 Chelyabinsk, Russia; vadimed@yandex.ru (V.T.); mkomelkova@mail.ru (M.K.); manukh@mail.ru (E.M.); freddowney@yahoo.com (H.F.D.); f.schwarcz@yandex.ru (P.P.); pashkov-anton@mail.ru (A.P.); diol2008@yandex.ru (O.T.); maltsevanv@susu.ru (N.M.); chrousos@gmail.com (G.P.C.); 2Institute of Immunology and Physiology, Ural Branch of the Russian Academy of Science, 620049 Ekaterinburg, Russia; v.chereshnev@iip.uran.ru (V.C.); mchereshneva@mail.ru (M.C.); 3Faculty of Medicine, Chelyabinsk State University, 454001 Chelyabinsk, Russia; 4Laboratory for Immunomorphology of Inflammation, Research Institute of Human Morphology, 117418 Moscow, Russia; actual_probl@mail.ru; 5Laboratory for Regulatory Mechanisms of Stress and Adaptation, Institute of General Pathology and Pathophysiology, 125315 Moscow, Russia; goryacheva@mail.ru; 6Department of Physiology and Anatomy, University of North Texas Health Science Center, Fort Worth, TX 76107, USA; 7Laboratory of Neuroendocrinology, I.P. Pavlov Institute of Physiology RAS, 6 Emb. Makarova, 199034 Saint Petersburg, Russia; julia.fedotova@mail.ru; 8International Research Centre “Biotechnologies of the Third Millennium”, ITMO University, 191002 Saint Petersburg, Russia; 9Biophysics Laboratory, Institute of Laser Physics, Siberian Branch of the Russian Academy of Science, 630090 Novosibirsk, Russia; cherk_63@mail.ru; 10Department of Medicine, Technical University of Dresden, 01309 Dresden, Germany; charlotte.steenblock@uniklinikum-dresden.de (C.S.); stefan.bornstein@uniklinikum-dresden.de (S.R.B.); 11Rayne Institute, Division of Diabetes & Nutritional Sciences, Endocrinology and Diabetes, Faculty of Life Sciences & Medicine, Kings College London, London SE5 9PJ, UK; 12Department of Psychiatry and Psychotherapy, University of Leipzig, 04107 Leipzig, Germany; barbara.ettrich@googlemail.com; 13University Research Institute of Maternal and Child Health and Precision Medicine, National and Kapodistrian University of Athens, 11527 Athens, Greece; 14Department of Child and Adolescent Psychiatry, Psychotherapy, and Psychosomatics, University of Leipzig, 04107 Leipzig, Germany

**Keywords:** CPTSD, glucocorticoids, adrenal gland, behavior

## Abstract

Background: Rats exposed to chronic predator scent stress mimic the phenotype of complex post-traumatic stress disorder (PTSD) in humans, including altered adrenal morphology and function. High- and low-anxiety phenotypes have been described in rats exposed to predator scent stress (PSS). This study aimed to determine whether these high- and low-anxiety phenotypes correlate with changes in adrenal histomorphology and corticosteroid production. Methods: Rats were exposed to PSS for ten days. Thirty days later, the rats’ anxiety index (AI) was assessed with an elevated plus-maze test. Based on differences in AI, the rats were segregated into low- (AI ≤ 0.8, n = 9) and high- (AI > 0.8, n = 10) anxiety phenotypes. Plasma corticosterone (CORT) concentrations were measured by ELISA. Adrenal CORT, desoxyCORT, and 11-dehydroCORT were measured by high-performance liquid chromatography. After staining with hematoxylin and eosin, adrenal histomorphometric changes were evaluated by measuring the thickness of the functional zones of the adrenal cortex. Results: Decreased plasma CORT concentrations, as well as decreased adrenal CORT, desoxyCORT and 11-dehydroCORT concentrations, were observed in high- but not in low-anxiety phenotypes. These decreases were associated with increases in AI. PSS led to a significant decrease in the thickness of the *zona fasciculata* and an increase in the thickness of the *zona intermedia*. The increase in the thickness of the *zona intermedia* was more pronounced in low-anxiety than in high-anxiety rats. A decrease in the adrenal capsule thickness was observed only in low-anxiety rats. The nucleus diameter of cells in the *zona fasciculata* of high-anxiety rats was significantly smaller than that of control or low-anxiety rats. Conclusion: Phenotype-associated changes in adrenal function and histomorphology were observed in a rat model of complex post-traumatic stress disorder.

## 1. Introduction

The adrenal gland plays a major role in response to physiological challenges and is able to adapt to homeostatic alterations. Proper adaptation is of particular importance since dysregulation of the endocrine stress system may be the cause of various mental diseases [1]. For example, impaired glucocorticoid (GC) signaling may play a significant role in the pathogenesis of stress-related disorders, while lower plasma GC levels were found in certain individuals with chronic stress [2,3]. In several disorders, including atypical depression, fibromyalgia and chronic fatigue syndrome, neuroendocrine changes were related to impaired GC signaling [4]. Furthermore, clinical and experimental data have indicated decreased plasma, salivary, and urinary GC concentrations in post-traumatic stress disorder (PTSD). Particularly, individuals with low cortisol concentrations after acute trauma had significantly higher risk of subsequently developing PTSD symptoms [5]. Interestingly, treatment with hydrocorticosterone (hydroCORT) reduced the risk of PTSD. Notwithstanding the evidence supporting GC insufficiency in PTSD, the pathophysiological mechanisms of the behavioral disorders remain unknown. In fact, PTSD-related neuroinflammation might be a probable mediator of GC signal insufficiency [6].

Reduced GC secretion and/or action goes along with adrenal insufficiency as has been shown in clinical and experimental studies [7]. Furthermore, we previously reported that chronic predator scent stress (PSS)-induced PTSD was accompanied by adrenal damage, especially in the GC-producing zona fasciculata, as well as plasma corticosterone (CORT) reduction [8]. Moreover, we found inverse CORT levels in high- vs. low-anxiety allostatic flight/fight/active (AFR) or allostatic freezing/passive (APR) responders to chronic stress [9]. However, no histomorphological data related to CORT reduction in chronically stressed subjects are available. Our animal model allows filling this lack of evidence, as segregation of all stress-exposed phenotypes related to PTSD intensity is possible. Thus, we correlated histomorphological alterations in the adrenals with changes in plasma and adrenal CORT concentrations in high- and low-anxiety rats after exposure to PSS.

## 2. Results

### 2.1. PSS Affects the Anxiety of Rats in an Elevated Plus-Maze Test

To assess the effect of different anxiety phenotypes on the adrenal cortex, we exposed rats to PSS for 10 days. After 30 days of restoration, we performed an elevated plus-maze (EPM) test. Mean EPM test values (Table 1) of PSS rats did not differ from those of control rats. However, significant differences were detected when PSS rats were separated into behavioral phenotypes. High-anxiety rats spent more time in the closed arms of the EPM (*p* = 0.025) and less time in the open arms (*p* = 0.025). In addition, high-anxiety rats entered the closed arms more frequently (*p* = 0.0002). The number of entries into the closed arms of the EPM by low-anxiety rats was 67% lower than that of control rats (*p* = 0.0008) and 75% lower than that of high-anxiety rats (*p* = 0.0002). The anxiety index (AI) of high-anxiety rats exceeded that of control rats (*p* = 0.0021), whereas the AI of low-anxiety rats did not differ significantly from that of control rats.

### 2.2. PSS Causes a Decrease in Plasma CORT Concentrations in High-Anxiety Rats

Plasma CORT concentrations (Figure 1A) in PSS rats did not differ from control values (*p* = 0.06), while plasma CORT concentrations in high-anxiety rats were 53% lower than in control rats (*p* < 0.01) and 60% lower than in low-anxiety rats (*p* < 0.001). Plasma CORT concentrations of low-anxiety and control rats did not differ significantly. For all PSS rats, plasma CORT concentrations and AI were negatively correlated (r = −0.79, *p* = 0.00059; Figure 1B).

### 2.3. PSS Causes a Decrease in Adrenal Corticosteroids

Adrenal CORT concentrations (Figure 1C) in PSS rats did not differ from control values (*p* = 0.1). In high-anxiety rats, adrenal CORT concentration was 62% lower than in control rats (*p* < 0.001) and 67% lower than in low-anxiety rats (*p* < 0.001). For all PSS rats, plasma CORT concentrations and AI were negatively correlated (r = −0.54, *p* = 0.02; Figure 1D).

Adrenal desoxyCORT concentrations (Figure 2A) in PSS rats were 26% lower than in control rats (*p* < 0.05). In high-anxiety rats, desoxyCORT was 46% lower than in control rats (*p* < 0.001) and 50% lower than in low-anxiety rats (*p* < 0.001). For all PSS rats, plasma desoxyCORT concentrations and AI were negatively correlated (r = −0.5, *p* = 0.03; Figure 2B).

Adrenal 11-dehydroCORT concentrations (Figure 2C) of PSS rats did not differ from those of control rats. High-anxiety rats had 43% lower dehydroCORT concentrations than control rats (*p* < 0.01) and 53% lower concentrations than low-anxiety rats (*p* > 0.01). Adrenal 11-dehydroCORT concentrations of low-anxiety rats did not differ from those of control rats. Unlike adrenal CORT, there were no significant correlations between adrenal 11-dehydroCORT concentrations and AI (Figure 2D).

### 2.4. PSS Affects the Thickness of the Adrenal Cortex

In control rats, the cell cytoplasm of the *zona glomerulosa* and especially the *zona fasciculata*, contained a large amount of sudanophilic lipid inclusions. The *zona reticularis* consists of small adrenocorticocytes that compose retiform networks of cords. Figure 3 shows that in high-anxiety rats, numerous adrenocorticocytes of the *zona glomerulosa* and especially the *zona fasciculata* displayed a pronounced decrease in density with lipid inclusions and delipidation of the cytoplasm. In low-anxiety rats, the delipidation was visible in only a few cells.

The thickness of the entire adrenal cortex (Figure 4) was decreased in PSS rats (*p* < 0.01). The adrenal cortices of high- and low-anxiety rats were ~25% thinner than that of control rats (*p* < 0.01); these values did not differ significantly.

The thickness of the capsule (Figure 5) was decreased by 17% in PSS rats (*p* < 0.05) compared to control rats. However, in high-anxiety rats, this difference was not significant. The capsule in low-anxiety rats was 17% thinner than in control rats (*p* < 0.01) and 20% thinner than in high-anxiety rats (*p* < 0.05).

Changes in the thickness of the *zona glomerulosa* were not significant among the groups. The thickness of the *zona intermedia* was increased (59%) in PSS rats compared to control rats (*p* < 0.001; Figure 6A). Likewise, the *zona intermedia* of high- and low-anxiety rats was increased (39%; *p* < 0.001 and 81%; *p* < 0.001, respectively) compared to the *zona intermedia* of control rats. This increase in low-anxiety rats was greater than in high-anxiety rats (*p* < 0.01). AI correlated negatively with the thickness of the *zona intermedia* (r = −0.46, *p* = 0.045; Figure 6B).

There was a positive correlation between adrenal 11-desoxyCORT concentration and *zona intermedia* thickness in the PSS group. (r = −0.56, *p* = 0.01, Figure 6C).

The thickness of the *zona fasciculata* (Figure 7) was 40% lower in PSS rats than in control rats (*p* < 0.001). Likewise, the *zona fasciculata* was 36% and 31% thinner in high- and low-anxiety rats, respectively (*p* < 0.001), for both comparisons. The zone thickness of high- vs. low-anxiety groups did not differ significantly.

### 2.5. PSS Affects Nucleus Diameter in the Zona Fasciculata

The average nucleus diameter in the *zona fasciculata* of PSS rats (Figure 8) was 40% smaller than that of control rats (*p* < 0.05). The average *zona fasciculata* nucleus diameter of high-anxiety rats was 11% smaller than in control rats (*p* < 0.01) and 12% smaller than in low-anxiety rats (*p* < 0.001). The nucleus diameter of low-anxiety rats did not differ significantly from that of control rats.

## 3. Discussion

In this study, an extended, experimental model of chronic stress/PTSD was used. In a previous study [8], we assessed behavioral activity on day 14 after exposure to PSS. Now, the post-stress rest time was increased to 30 days for estimation of more long-lasting effects of PSS than in earlier studies. Furthermore, PSS rats were segregated into low- and high-anxiety phenotypes based on their performance in an EPM test. These results confirm the findings of previous studies using this rat model of PTSD [10]. Plasma CORT concentrations were reduced and associated with hypoplastic adrenals, even when anxiety phenotypes were not considered. Previously, it was suggested that this CORT reduction is a consequence of the freezing-inducing pheromone 2,5-dihydro-2,4,5-trimethylthiazoline leading to a damped LHPA axis [10]. However, the present findings indicate that this conclusion must be revised, granted that in low-anxiety rats, plasma CORT concentrations were completely restored. Hence, CORT reduction in the plasma of low-anxiety rats was transient. Conversely, high-anxiety rats demonstrated CORT reduction both two weeks and one month after a sustained stress exposure. Therefore, we suggest that prolonged rather than transient hypocortisolemia is involved in the development of anxiety symptoms in human PTSD. Similarly to the current study, Skórzewska and colleagues found that PTSD-susceptible rats, compared to PTSD-resilient rats, had decreased concentrations of CORT in plasma, as well as reduced corticotropin-releasing factor expression in the paraventricular nuclei in a stress–restress paradigm [11].

We found that hypocorticosteronemia in high-anxiety rats was associated with a decrease in the CORT concentration in the adrenals. A simultaneous decrease in 11-desoxycorticosterone concentration indicates a suppression of steroidogenesis in the adrenals of these high-anxiety rats. In contrast, in low-anxiety rats, adrenal GC production was unchanged in comparison to that of controls, which may prevent prolonged hypocorticosteronemia development. In high-anxiety rats, a reduction in adrenal GC synthesis, which is synchronous with the decrease in 11-dehydrocorticosterone concentration, indicates a reduction in 11-beta-hydroxysteroid dehydrogenase (11-beta-HSD) activity, which, in turn, catalyzes the conversion of CORT into 11-dehydroCORT. The negative correlations between AI and plasma CORT concentrations and between AI and adrenal desoxyCORT concentrations possibly reflect the involvement of GCs in the down-regulation of anxiety-like PTSD symptoms.

There are two isoforms of 11-beta-HSD, a low-affinity NADP(H)-dependent dehydrogenase/ 11-oxoreductase (11-beta-HSDl) and a high-affinity NAD-dependent dehydrogenase (11-beta-HSD2). The adrenal glands of rats express both 11-beta-HSDl and 11-beta-HSD2; however, the expression of 11-beta-HSD2 prevails on a functional level [12]. ACTH induces 11-beta-HSD2 activity in cells isolated from the *zona fasciculata* of rats [13]. Thus, in high-anxiety rats, simultaneous reduction in 11-dehydroCORT and 11-desoxyhyCORT concentrations might reflect the alterations of ACTH signaling in the adrenal glands. Others linked plasma ACTH reduction to a low sensitivity to ACTH of the *zona fasciculata* [14]. Further studies will focus on the experimental verification of these hypotheses with respect to high-anxiety rats.

The nucleus diameter of adrenocorticocytes was smaller in high-anxiety rats than in control rats, reflecting a relation between adrenal histology and chronic stress/PTSD. Moreover, a negative correlation between nucleus diameter in the *zona fasciculata* and time spent in the closed arms was observed in stressed rats. Further, there was a negative correlation between nucleus diameter in the *zona intermedia* and time spent in the closed arms in stressed rats. These results verify adrenal histology changes in chronic stress.

GCs are the mediators between PTSD-like anxiety symptoms and histological adrenal alterations, a relation that is supported by positive correlations between nucleus diameter in the *zona fasciculata* and CORT concentration in plasma and the adrenal glands, as well as between nucleus diameter in the *zona intermedia* and 11-desoxyCORT concentration in adrenals of PSS-exposed rats. Meanwhile, plasma and adrenal CORT concentrations, akin to 11-desoxyCORT and 11-dehydroCORT concentrations, exhibited negative correlations with AI in stressed rats.

Previously, we associated decreased blood CORT concentrations with histological abnormalities in the adrenal *zona fasciculata*, including appearance of degenerating, ballooning, and hydropic cells, two-weeks after PSS. In that study, we focused on the negative correlation between the anxiety conditions and thickness of the *zona fasciculata* as it confirmed the significance of adrenal gland dystrophy in the development of PTSD-related behaviors [8]. In this study, we observed in both phenotypes of chronically stressed rats, a *zona fasciculata* reduction, as well as a *zona intermedia* enlargement. Notably, the increase in the *zona intermedia* thickness in low-anxiety rats was more pronounced than that in high-anxiety rats. The negative correlation between the AI value and *zona intermedia* thickness in PSS-exposed rats confirms the significance of this zone in anxiety behavior regulation.

The *zona intermedia* is represented by immature cells and, therefore, considered as an adrenal stem and progenitor cell-enriched zone. There is a large amount of literature data suggesting that progenitor cells, activated by cytokines or growth factors in theirs niches, can migrate to their target location in tissues [15]. In the adrenal cortex of adult mice under normal circumstances, a number of different stem and progenitor cells initiate replacement of senescent cells by newly differentiated cells [16]. The best characterized cell population of the adrenal capsule is distinguished by the expression of GLI1, a transcriptional effector of the canonical Hedgehog pathway. In the adult adrenal cortex, contribution of capsular GLI1-positive cells to tissue renewal is relatively small though significantly higher in females than in males [17]. However, other studies have shown that these cells can still be activated in adult male mice when needed [18]. Other progenitor cells in the adrenal cortex seem to be quiescent under normal homeostatic conditions, but remain able to respond to specific physiological cues [19]. For example, WT1-positive progenitors, which are otherwise just active during embryonic development, expand after gonadectomy [20]. Furthermore, we have identified nestin-positive progenitors in the adult adrenal cortex, which under stress differentiate into steroidogenic cells and migrate centripetally through the adrenal cortex [21]. This suggests that special stress-responsive progenitor cells can give rise to morphological changes in the adrenal cortex [22], as we have also previously observed after metabolic stress [23]. Thereby, the increase in the *zona intermedia* thickness that we observed in PSS-exposed rats might be interpreted as a compensatory response aimed at the maintenance of *zona fasciculata* stability.

Results of the current study demonstrated that plasma and adrenal GC concentrations were higher in low- than in high-anxiety rats. These higher concentrations were associated with an increase in the *zona intermedia* thickness. In high-anxiety rats, these histologic alterations in the adrenal cortex were not sufficient to restore GC production. In general, these results are in a good agreement with previous data indicating that immature cells play a crucial role in adrenal plasticity [22]. As CORT production was not suppressed in low-anxiety rats, the observed increase in the *zona intermedia* thickness in these rats suggests that their resistance to stress may have been associated with an activation of adrenocortical stem or progenitor cells.

Although we found that CORT production was inhibited in the adrenal glands of high-anxiety rats, we can only speculate about the potential causes of this finding. Perhaps, a decrease in the concentration of ACTH and/or a diminution in the sensitivity of the *zona fasciculata* cells to ACTH is responsible for this finding, probably because of decreased brain signaling through the splanchnic nerve. This should be clarified in future studies using a chronic stress paradigm, and in patients with complex PTSD.

## 4. Materials and Methods

### 4.1. Experimental Animals

Experiments were performed on male Wistar rats, weighing 210–230 g and three months old at the beginning of this study. Thirty-nine rats were randomly divided into two groups: (1) control and (2) PSS. Rats were housed in standard cages and received rat chow and tap water ad libitum. The animals were kept at controlled temperature (22–25 °C) and humidity (55%). A 12:12 h light–dark cycle was maintained with lights on between 07:00 and 19:00. All animal procedures were performed in accordance with the U.S. National Research Council Guide for the Care and Use of Laboratory Animals (publication 85–23, revised 2011). The experimental protocols (Project 0520-2019-0030) were approved by the Animal Care and Use Committee of the Institute of General Pathology and Pathophysiology (18 January 2019).

### 4.2. Modeling PTSD

To induce PTSD, we used a modified model of predator stress that was initially described by Cohen and Zohar [24] and as used in our prior studies [8]. Predator stress was accomplished by exposing rats of the PSS group to cat urine scent for 10 min daily for 10 days. PSS rats were then were given 30 days rest under predator stress-free conditions. Control rats were rested during this 40 day period.

### 4.3. Behavioral Testing

The predator stress outcome was evaluated using the EPM test as employed in our prior study [8]. This test was performed on the day following completion of the post-stress, 30 day rest period (PSS group). The total duration of the test was 10 min. Control rats were tested together with rats from experimental groups in a blinded fashion. The behavior of rats in the maze was recorded and tracked using the video system SMART and analyzed with SMART 3.0 software. The number of entries into open and closed arms of the EPM and the time spent in the open and closed arm were recorded. Based on these measurements, an anxiety index (AI) was calculated [25]:AI = 1 − {[(time in open arms/Σtime on maze) + (number of entries into open arms/Σnumber of all entries)]/2}

### 4.4. Blood and Tissue Collection and Storage

Rats were sacrificed by an overdose of diethyl ether, decapitated, and blood was collected on experimental day 28. Blood serum was removed from clotted blood and stored in Eppendorf tubes at −70 °C. The adrenals were excised and stored in 10% buffered formalin for histological analysis or frozen in liquid nitrogen and stored at −70 °C for biochemical studies.

### 4.5. Plasma Corticosterone and Adrenal Corticosteroid Measurements

The entire procedure of adrenal CORT, desoxyCORT and 11-dehydroCORT extraction, measurements, and validation was previously described in detail [9]. In brief, the adrenal glands were weighed and thoroughly homogenized in 1 mL of cold acetone. The homogenates were centrifuged at 2000× *g* at 4 °C for 15 min. The supernatants were evaporated in a nitrogen flow at 40 °C. The residues were dissolved in 24 µL of a 65% methanol solution and 8 µL of solution was analyzed by high-performance liquid chromatography (HPLC) on a “Milichrom-1” chromatograph (Nauchpribor, Russia). Chromatographic conditions were as follows: steel column 2 × 62 mm in size, packed with Silasorb C18 SPH 8 (5 µm) as sorbent, gradient elution with acetonitrile in water from 30 to 55% (*v*/*v*), eluting rate of 100 µL/min, and UV detection at 240 and 260 nm. The amounts of the corticosteroids were determined in ng per mg of tissue (ng/mg). Chromatographic separation of a mixture of standards (A) and adrenal extracts (B) is shown in Figure 9.

### 4.6. Histomorphology

The left adrenal gland was fixed in buffered 10% formalin, embedded in paraffin, and 5 to 7 μm equatorial sections were made as previously described [8]. The sections were stained with hematoxylin and eosin. The *zona intermedia* located between the *zona glomerulosa* and *zona fasciculata* was verified by staining cryostat sections for fats and lipoids with Sudan III, since this zone is not stained with Sudan III. The sections were photographed with an Axioplan 2 imaging microscope and analyzed with a morphometrical program, AxioVision (Carl Zeiss Microscopy GmbH, Jena, Germany). On each section, a functional zone of the adrenal cortex, i.e., *zona intermedia* and *zona fasciculata*, was measured in at least 10 different random locations.

## 5. Data Analyses

Data were analyzed with SPSS 24 (IBM, New York, NY, USA), STATISTICA 10.0 (StatSoft, Tulsa, OK, USA), Rstudio (RStudio, Boston, MA, USA) and Excel (Microsoft, Redmond, WA, USA) software. The normality of data distributions was examined with the Shapiro–Wilk test. Data are presented as the mean ± SEM or as median (25th–75th percentile). Normally distributed data were analyzed with a parametric, one-factor ANOVA followed by Tukey’s post hoc tests to compare all outcome measures between respective groups, e.g., control vs. PSS, control vs. high-anxiety, control vs. low-anxiety, high-anxiety vs. low-anxiety. Non-normally distributed data were analyzed with a nonparametric, one factor Kruskal–Wallis ANOVA followed by Dunn tests for pairwise comparisons between respective groups. Relationships between variables were examined by Spearman correlation analysis. *p* < 0.05 was considered significant.

## 6. Conclusions

This study demonstrate phenotype-associated responses of adrenal glands in an animal model of complex PTSD. Blood, adrenal CORT and adrenal desoxyCORT concentrations of high-anxiety rats were decreased. Thus, we conclude that GC production is suppressed in this phenotype. Consistent with this conclusion, histological data showed more pronounced abnormalities in adrenals of high-anxiety rats as compared to low-anxiety rats.

## 7. Limitations of This Study

A limitation of our study is that we did not investigate the mechanism responsible for the observed differences between high- and low-anxiety rats. Furthermore, we only analyzed male rats although there is increasing evidence that sex differences play an important role in mental diseases.

## Figures and Tables

**Figure 1 ijms-22-13235-f001:**
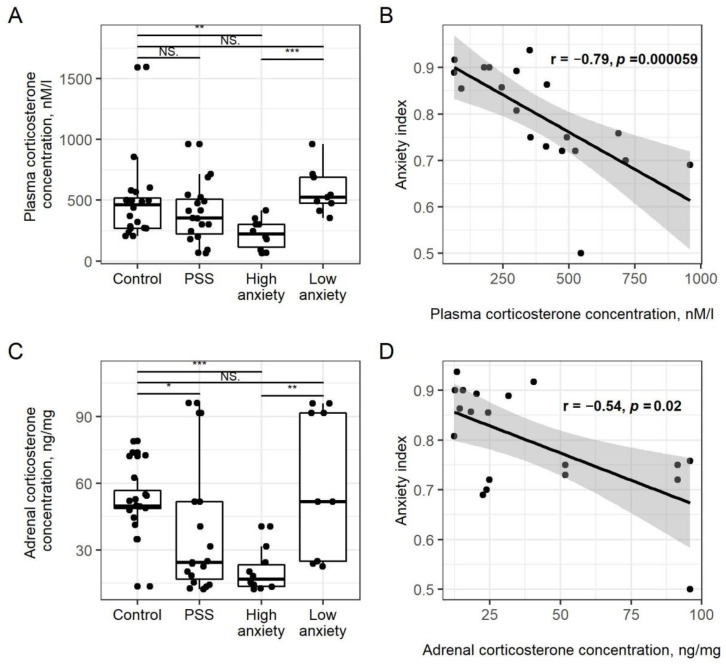
**PSS causes a decrease in corticosterone**. Panel (**A**). Plasma CORT concentrations in PSS rats segregated into high- and low-anxiety groups according to AI as determined by performance in the EPM. Differences in plasma CORT concentrations among groups are shown as boxplots with dots representing individual data values and medians shown by horizontal lines. The boxes include the central 50% of the data, i.e., from the 25th to the 75th percentile. The whiskers include the data contained within 1.5 times the interquartile range. NS = *p* > 0.05; * *p* < 0.05; ** *p* < 0.01; *** *p* < 0.001. *p* values determined in non-parametric analysis. Panel (**B**). Spearman correlation between AI values and plasma CORT concentrations in PSS rats. Gray area around the line represents the 95% confidence interval. Panel (**C**). Adrenal CORT concentrations. See Panel A legend for detailed description of the plot. Panel (**D**). Spearman correlation between AI and adrenal CORT concentrations in PS rats. Gray area around the line represents the 95% confidence interval.

**Figure 2 ijms-22-13235-f002:**
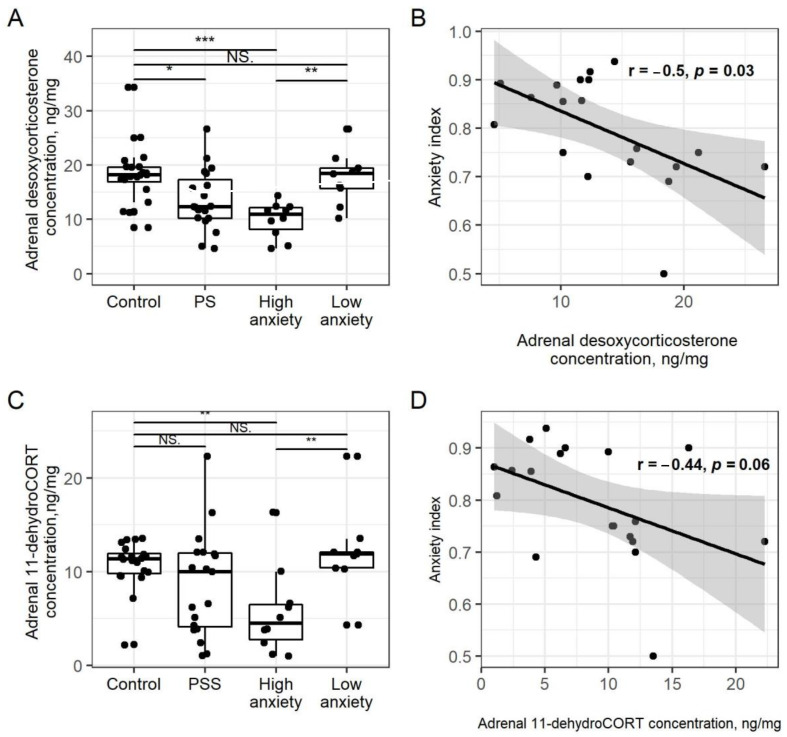
**PSS causes a decrease in desoxyCORT and 11**-**dehydroCORT**. Panel (**A**). Adrenal desoxyCORT concentrations. Differences in adrenal desoxyCORT concentrations among groups are shown as boxplots with dots rep-resenting individual data values and medians shown by horizontal lines. The boxes include the central 50% of the data, i.e., from the 25th to the 75th percentile. The whiskers include the data contained within 1.5 times the interquartile range. NS = *p* > 0.05; * *p* < 0.05; ** *p* < 0.01; *** *p* < 0.001. *p* values determined in non-parametric analysis. Panel (**B**). Spearman correlation between AI and adrenal desoxyCORT concentration in PSS rats. Gray area around the line represents the 95 % confidence interval. Panel (**C**). Differences in adrenal 11-dehydroCORT concentrations between groups. See Panel (**A**) legend for description of the plots. Panel (**D**). Spearman correlation between AI and adrenal 11-dehydroCORT concentration in PSS rats.

**Figure 3 ijms-22-13235-f003:**
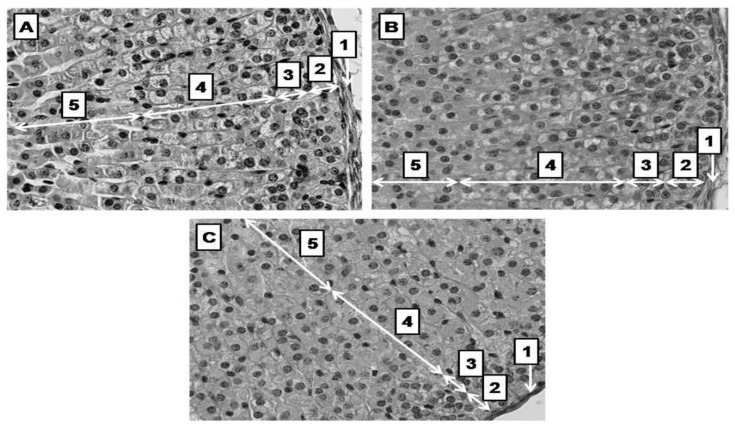
**Adrenal morphology is changed after PSS**. Adrenal cortex of high-anxiety (**A**), low-anxiety (**B**), and control (**C**) rats. 1, capsule; 2, *zona glomerulosa*; 3, *zona intermedia*; 4, *zona fasciculata*; 5, *zona reticularis*.

**Figure 4 ijms-22-13235-f004:**
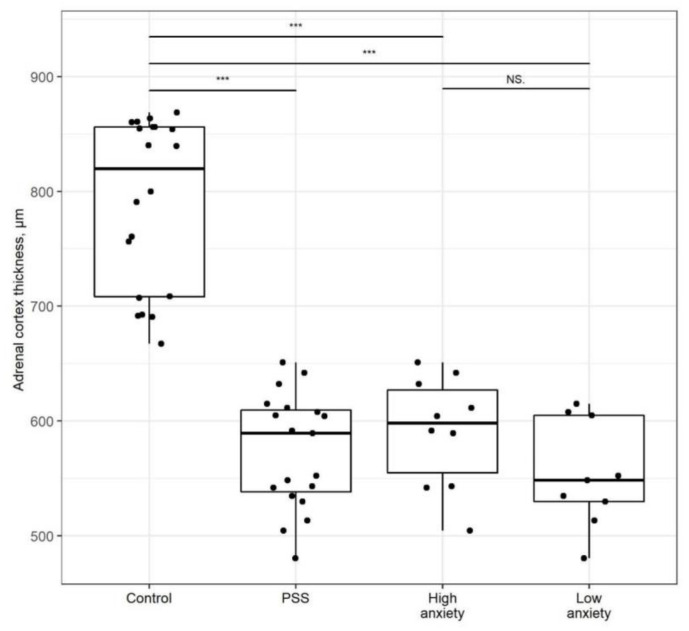
**Adrenal cortex thickness is decreased after PSS**. Adrenal cortex thickness of control, PSS, high- and low-anxiety PSS groups. Differences in adrenal cortex thickness values among groups are shown as boxplots with dots rep-resenting individual data values and medians shown by horizontal lines. The boxes include the central 50% of the data, i.e., from the 25th to the 75th percentile. The whiskers include the data contained within 1.5 times the interquartile range. NS = *p* > 0.05; *** *p* < 0.001. *p* values determined in non-parametric analysis.

**Figure 5 ijms-22-13235-f005:**
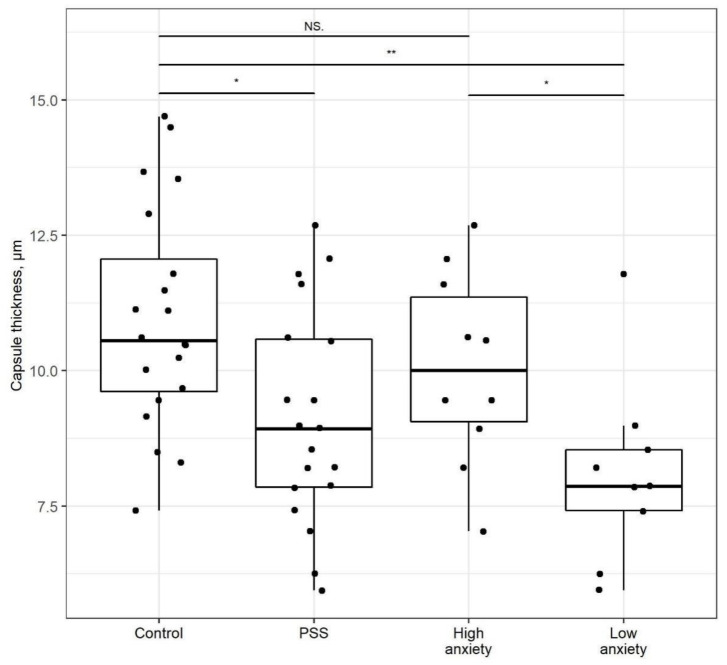
**Adrenal capsule thickness is changed after PSS**. Capsule thickness of control, PSS, high- and low-anxiety groups. Differences in adrenal cortex thickness values among groups are shown as boxplots with dots rep-resenting individual data values and medians shown by horizontal lines. The boxes include the central 50% of the data, i.e., from the 25th to the 75th percentile. The whiskers include the data contained within 1.5 times the interquartile range. NS = *p* > 0.05; * *p* < 0.05; ** *p* < 0.01. *p* values determined in parametric analysis.

**Figure 6 ijms-22-13235-f006:**
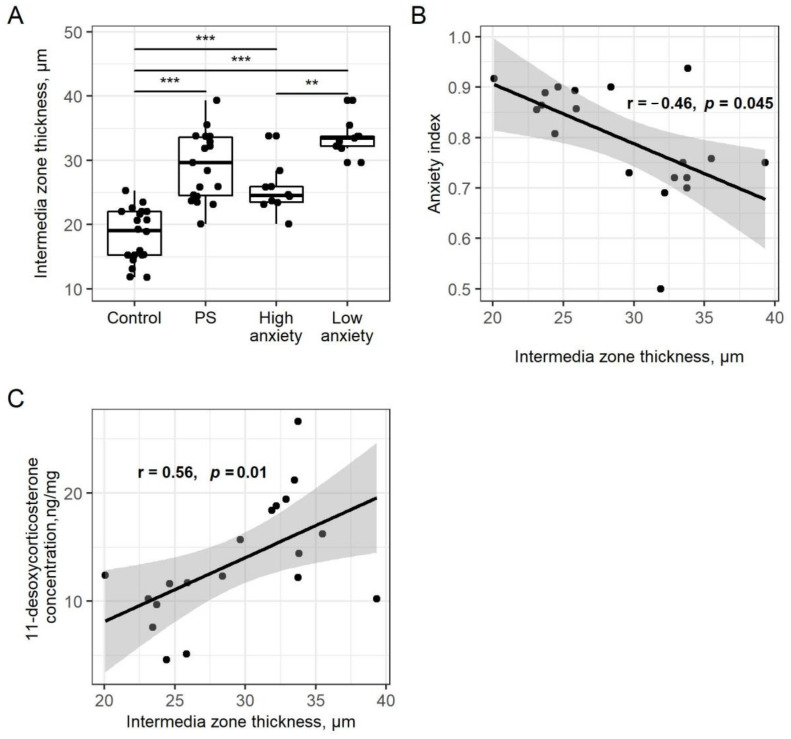
Panel (**A**). **Adrenal zona intermedia thickness is increased after PSS**. *Zona intermedia* thickness of control, PSS, high- and low-anxiety groups. Differences in adrenal cortex thickness values among groups are shown as boxplots with dots rep-resenting individual data values and medians shown by horizontal lines. The boxes include the central 50% of the data, i.e., from the 25th to the 75th percentile. The whiskers include the data contained within 1.5 times the interquartile range. NS = *p* > 0.05; ** *p* < 0.01; *** *p* < 0.001. *p* values determined in parametric analysis. Panel (**B**). Correlation between AI values and *zona intermedia* thickness in the PSS group (r = −0.46, *p* < 0.045). The gray area around the line represents the 95% confidence interval. Panel (**C**). Correlation between adrenal 11-desoxyCORT concentration and *zona intermedia* thickness in the PSS group (r = −0.56, *p* = 0.01). The gray area around the line represents the 95% confidence interval.

**Figure 7 ijms-22-13235-f007:**
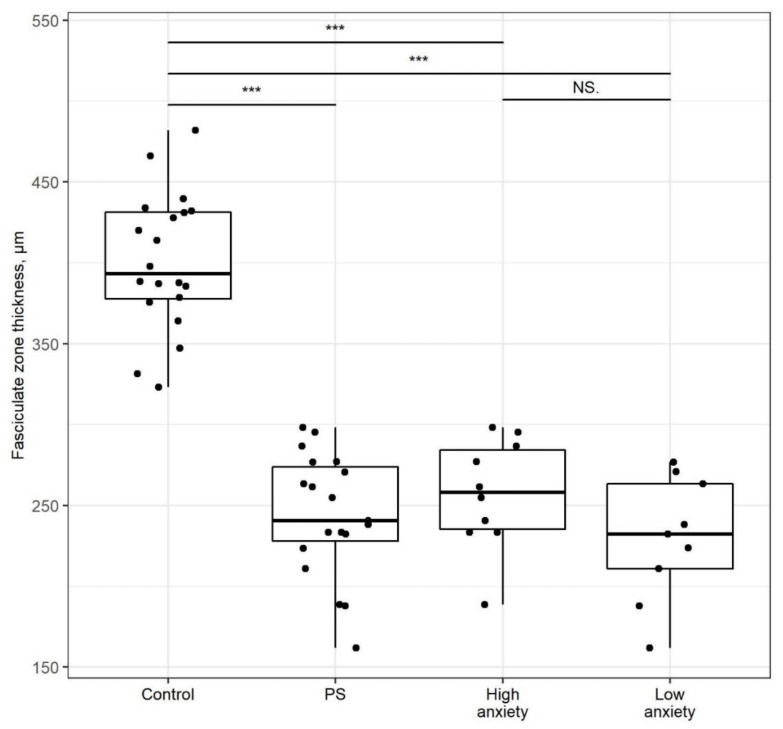
**Adrenal *zona fasciculata* thickness is decreased after PSS**. *Zona fasciculata* thickness of control, PSS, high- and low-anxiety groups. Differences in adrenal cortex thickness values among groups are shown as boxplots with dots rep-resenting individual data values and medians shown by horizontal lines. The boxes include the central 50% of the data, i.e., from the 25th to the 75th percentile. The whiskers include the data contained within 1.5 times the interquartile range. NS = *p* > 0.05; *** *p* < 0.001. *p* values determined in parametric analysis.

**Figure 8 ijms-22-13235-f008:**
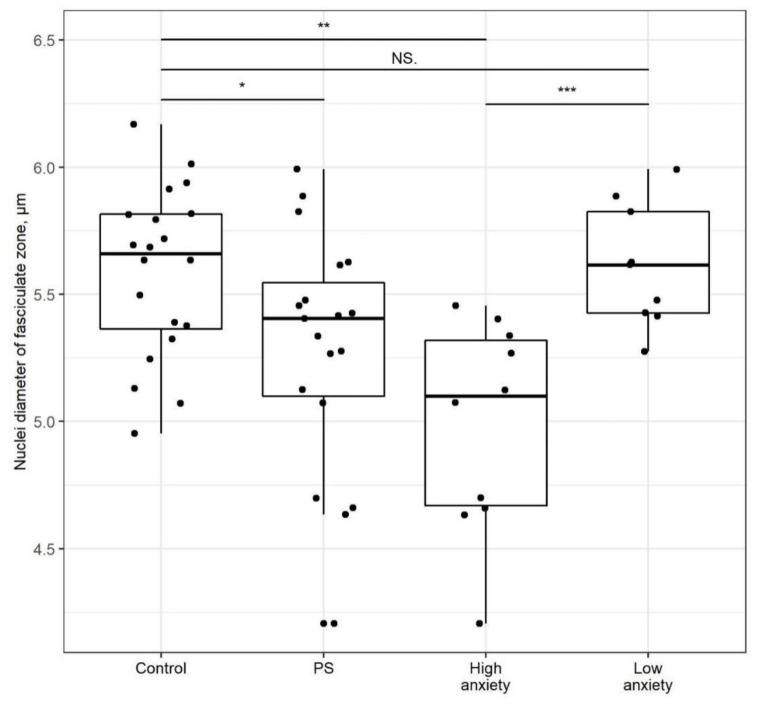
**Nucleus diameter in the *zona fasciculate* is decreases after PSS**. Nucleus diameter of cells in the *zona fasciculata* of control, PSS, high- and low-anxiety rats. Differences in adrenal cortex thickness values among groups are shown as boxplots with dots rep-resenting individual data values and medians shown by horizontal lines. The boxes include the central 50% of the data, i.e., from the 25th to the 75th percentile. The whiskers include the data contained within 1.5 times the interquartile range. NS = *p* > 0.05; * *p* < 0.05; ** *p* < 0.01; ****p* < 0.001. *p* values determined in parametric analysis.

**Figure 9 ijms-22-13235-f009:**
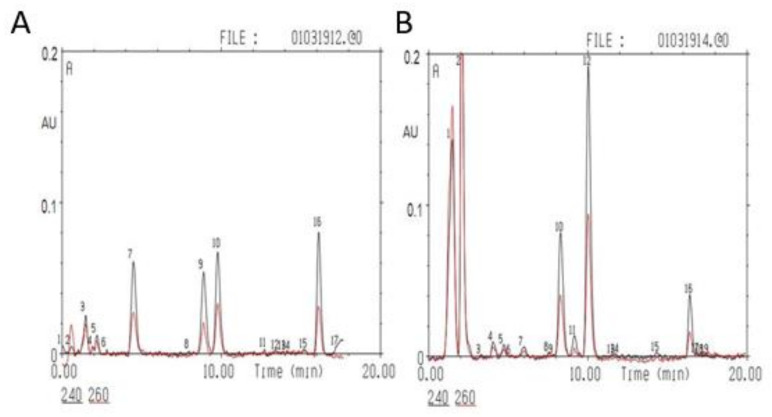
**Steroid analysis**. Chromatographic separation of a mixture of standards: (**A**). 7–aldosterone, 911-dehydroCORT, 10–CORT, and 16–desoxyCORT (C = 10 ng/µL, 5 μL was taken for the analysis); and adrenal extracts (**B**). 5–aldosterone, 11–11-dehydroCORT, 12–CORT, and 16–desoxyCORT. The black line corresponds to absorption at 240 nm, the red line to absorption at 260 nm. The y-axis represents absorption (A) in absorption units (AU); the x-axis shows time in min.

**Table 1 ijms-22-13235-t001:** Results of EPM behavioral experiments.

	Control (*n* = 20)	PSS (*n* = 19)	High Anxiety (*n* = 10)	Low Anxiety (*n* = 9)
Time spent in open arms (s)	115 ± 14	89 ± 12	43 ± 7 *	108 ± 20 ###
Time spent in closed arms (s)	485 ± 14	511 ± 14	557 ± 7 *	492 ± 20 ###
Entries to open arms	3.3 ± 0.4	2.7 ± 0.2	2.4 ± 0.4	2.7 ± 0.4
Entries to closed arms	6.9 ± 0.7	6.6 ± 0.6	8.0 ± 0.7	2.7 ± 0.4 *** ###
AI	0.74 ± 0.02	0.78 ± 0.01	0.85 ± 0.03 **	0.66 ± 0.03 ###

Data are the mean ± SEM. different from control * *p* < 0.05, ** *p* < 0.01, *** *p* < 0.005. different from high-anxiety phenotype, ### *p* < 0.001. AIs of high- and low-anxiety rats differed a priori and were not compared statistically.

## Data Availability

Data available on request from the authors.

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
