# Peer review of "A Rat Model of Post-Traumatic Stress Syndrome Causes Phenotype-Associated Morphological Changes and Hypofunction of the Adrenal Gland"

_ijms, 2021, doi:10.3390/ijms222413235_

Round 1

Reviewer 1 Report

I thank the authors and the editors for the opportunity to review this interesting manuscript. 

This is generally a well-written paper. The methods are well described. The figures are clear and appropriate. 

My primary concern relates to the face validity of the underlying stress model used to induce the results which are the subject of this analysis, and the authors' use of the term "PTSD" to describe the resultant phenotype in rats. 

The authors’ modified stress model implies that rats exposed to “predator stress”, namely cat odor over the course of 10 days, will be at risk of developing a rat-specific version of PTSD, which the authors somewhat inappropriately call “PTSD”. However, my understanding is that rats are ubiquitously exposed to predators in nature.

Certainly, the experience of any graduate student living in blighted inner city housing will confirm that where there are rats, there is the odor of cat urine. Indeed, it would be be difficult to imagine a rat in the modern urban environment not exposed to cat odor. If the authors' model is to be considered valid, based on their results, one might expect roughly half of all such rats to develop such murine "PTSD". 

It is difficult at face value to accept that a phenotype observed in roughly half of rats exposed to ordinary living conditions can be equated with the deeply pathological condition we describe as PTSD. On what is here our Veterans Day, it is appropriate for me to comment that it seems trivializing and somewhat disparaging to use the term "PTSD" to describe this condition. 

While I recognize that these considerations have been somewhat considered elsewhere, it would be helpful if the authors could better justify their use of this model beyond merely citing this prior work.

Reviewer 2 Report

The Authors of the study described their research in detail. However, there are minor shortcomings in the work. Below is a list of them:

  1. Each description of the Figure corresponds to the description of Figure 1. This means that when reading the description in Figure 8, I have to scroll through the entire work to go back to the first drawing. This is a major inconvenience and should be changed.
  2. anxiety index (AI) - at least the abbreviation should appear in the text where it was first mentioned. The explanation of the abbreviation appears in the Abstract, and then only in the description of the method, i.e. at the end of the work.
  3. The samples after evaporation were dissolved in 65% methanol and this solution was injected onto the chromatography system. Why was not a solvent similar to the initial concentration of the mobile phase chosen in this case?
  4. All data on the determined concentrations are presented in the form of diagrams only. Presenting them also numerically in the form of tables would bring a lot to the work.
  5. The work also lacks chromatograms presenting the analyzed compounds.
